# Effects of Sacubitril/Valsartan on the Renal Resistance Index

**DOI:** 10.3390/jcm11133683

**Published:** 2022-06-26

**Authors:** Margherita Ilaria Gioia, Giuseppe Parisi, Dario Grande, Miriam Albanese, Gianmarco Alcidi, Michele Correale, Natale Daniele Brunetti, Marco Matteo Ciccone, Massimo Iacoviello

**Affiliations:** 1Cardiology Unit, Perrino Hospital, 72100 Brindisi, Italy; 2School of Cardiology, University of Bari, 70124 Bari, Italy; giuseppeparisi88@libero.it (G.P.); dario.grande@ymail.com (D.G.); albanesemiriam91@gmail.com (M.A.); marcomatteo.ciccone@gmail.com (M.M.C.); 3Local Health Agency of Bari, 70124 Bari, Italy; gianmarco.alcidi@gmail.com; 4Department of Medical and Surgical Sciences, University of Foggia, 71100 Foggia, Italy; michele.correale@libero.it (M.C.); natale.brunetti@unifg.it (N.D.B.); 5Cardiology Unit, Polyclinic University Hospital of Foggia, 71100 Foggia, Italy; 6Cardiology Unit, Polyclinic University Hospital of Bari, 70100 Bari, Italy

**Keywords:** heart failure with reduced ejection fraction, cardiorenal syndrome, angiotensin receptor–neprilysin inhibitors, reverse cardiac remodeling, renal resistance index

## Abstract

Background: Sacubitril/valsartan plays a key role in improving left ventricular remodeling and prognosis in patients with heart failure with a reduced ejection fraction (HFrEF). Moreover, some data support its role in preserving renal function. In order to better clarify the effects of sacubitril/valsartan in cardiorenal syndrome, this study evaluated its effects on the renal resistance index (RRI). Methods: A group of patients with HFrEF was enrolled. The RRI was assessed with renal echo-color Doppler at enrollment and again after at least six months of sacubitril/valsartan treatment. In a subgroup of patients, the RRI was also evaluated at least six months before enrollment. The variations in echocardiographic parameters reflecting the left and right ventricular function, as well as creatinine and the estimated glomerular filtration rate, were also evaluated. Results: After treatment with sacubitril/valsartan, significant improvements in the left ventricular ejection fraction, and a decrease in the left atrial and ventricular volumes were observed. The RRI also showed a significant decrease. No relationship was found between the improvements in the parameters reflecting cardiac function and changes in the RRI. Conclusions: Treatment with sacubitril/valsartan is associated with improvements in both left ventricular function and renal perfusion, through decreasing the renal resistance. These data help to clarify the effects of the drug on cardiorenal syndrome progression.

## 1. Introduction

In patients with heart failure with a reduced left ventricular ejection fraction (HFrEF), sacubitril/valsartan has been demonstrated to be superior to ACE inhibitors (ACEi) in decreasing the risk of heart failure hospitalization and death [1], and in reversing left ventricular remodeling [2,3]. These effects are associated with better neuro-hormonal modulation that is mediated by this drug, which both antagonizes angiotensin II and inhibits the degradation of natriuretic peptides (NPs) [4,5]. 

Some data have also demonstrated the ability of sacubitril/valsartan to slow the progression of renal dysfunction [6,7,8,9]. A secondary analysis of PARADIGM-HF [6] has indicated that, during follow-up studies, the patients taking sacubitril/valsartan had a smaller decrease in the estimated glomerular filtration rate (GFR) than the patients taking enalapril, despite showing a greater blood pressure decrease. These effects were independent of both chronic kidney disease and albuminuria. These findings have been further supported by the PARAMOUNT study and available metanalyses [7,8,9]. The mechanisms underlying these favorable effects on renal function have not been fully elucidated but may be mediated by improvements in the renal blood flow, which are mediated by the increased efficiency of the NP system (NPS) [5,10].

In order to better clarify the effects on the renal blood flow, the aim of this study was to evaluate the variation in the renal resistance index (RRI) after treatment with sacubitril/valsartan.

## 2. Materials and Methods

We evaluated patients referred to the Heart Failure Unit of the University Policlinic Hospital of Bari from 2016 and 2019, and to the Heart Failure Unit of the University Policlinic Hospital of Foggia from 2019 and 2021 for HFrEF (ESC criteria), who had been prescribed sacubitril/valsartan. Patients from Bari were enrolled in a study aimed at evaluating the predictors of cardiorenal syndrome progression whose main results have already been published [11]. Patients from Foggia were enrolled in the Daunia registry. Both of these studies were approved by local ethics committees and all enrolled patients provided written informed consent to participate.

Study design. Patients for whom sacubitril/valsartan was prescribed were evaluated. According to the indications of the Italian Ministry of Health, sacubitril/valsartan was prescribed to the patients with the New York Heart Association (NYHA) class II–IV; left ventricular ejection fraction (LVEF) of ≤ 35%; prior treatment with ACEi or angiotensin II receptor blockers (ARB) for at least 6 months; no history of angioedema; systolic arterial pressure of >95 mm Hg; estimated GFR of >30 mL/min/1.73 m^2^; and serum potassium of <5.2 mmol/L. Patients taking an ACEi before study enrollment underwent a 36 h washout before the start of treatment with sacubitril/valsartan. The starting dose of sacubitril/valsartan was 24/26 mg b.i.d. or 49/51 mg b.i.d. depending on arterial pressure, renal function, and the previous ARNi/ARB dose. The dose was then up-titrated, when tolerated, to 97/103 mg b.i.d.

Baseline evaluation (T0) was considered to be the time in which the sacubitril/valsartan therapy was started. Between 6 and 12 months after beginning ARNI, a new complete evaluation was performed (T1). Moreover, an evaluation was performed at 6 and 12 months before ARNI therapy was started (T-1). 

At T-1, T0, and T1, the following evaluations were performed:-Medical examination and ECG. Records were documented, including ischemic heart disease, arterial hypertension, diabetes mellitus, history of ventricular arrhythmic events, NYHA class, arterial pressure, heart rhythm and heart rate at ECG;-Echocardiographic examinations. Left ventricular end-diastolic volume (LVEDV), end-systolic volume (LVESV), and LVEF were calculated with Simpson’s rule. The peak of the E wave (E), through mitral pulsed Doppler at the level of the mitral leaflets, and early diastolic velocity peak (e’) at the level of the septal and lateral mitral annulus, through tissue Doppler imaging, were measured. The E/e’ ratio was then calculated as the ratio between E and the mean value of septal and lateral e’. The central venous pressure was determined through the assessment of the inferior vena cava diameter and respiratory excursion. The mitral regurgitation (MR) was evaluated and quantified in arbitrary units (a.u. range from 0 to 4). The systolic pulmonary artery pressure (PAP) was estimated by the measurement of the RV–right atrium gradient from the peak velocity of the tricuspid valve regurgitation (TR) with the simplified Bernoulli equation; this value was added to an estimate of the mean right atrium pressure. The RV systolic function was evaluated according to tricuspid annular plane systolic excursion (TAPSE);-Doppler of interlobular renal arteries. The method to assess the RRI was described previously [12,13]. The renal arterial Doppler was performed after echocardiographic examination by using the same echograph (Vivid 7, GE Vingmed Ultrasound, General Electric or EPIQ CVx system, Philips, Amsterdam, The Netherlands) and the same 4 MHz probe, moving the patient into the sitting position and using a posterior approach to the kidney. The course of the right or left kidney segmental arteries was visualized by color Doppler flow and then, at the middle tract level of the best visualized one, pulsed Doppler was performed. Every effort was made to achieve the best alignment of the ultrasonic beam. An average of 2–3 measurements of the peak systolic velocity and the end-diastolic velocity were used to calculate the RRI according to Peurcelot’s formula, i.e., 100 × [1 − (end-diastolic velocity/peak systolic velocity)].

RRI is a parameter with a high inter-operator and intra-operator reproducibility, as previously demonstrated [12]. To avoid bias in the measurement, the images were acquired by a single operator (M.I.) and analyzed by a single operator for each center (M.I.G. for patients referred to the center of Bari, and G.A. for those referred to the center of Foggia).

-Blood sample analyses. Blood samples were collected to evaluate NT-proBNP (immunoassay Dade Behring, Eschborn, Germany) and creatinine (mg/dL). The glomerular filtration rate was calculated with the abbreviated CKD-EPI formula (GFR-EPI, ml/min/1.73 m^2^) [14].

Study end-points. The primary end-point of the study was to evaluate the changes in the RRI between the evaluations before and the evaluation after the introduction of sacubitril/valsartan therapy. As the secondary end-point we evaluated the changes in echocardiographic parameters after sacubitril/valsartan therapy and their relationship with those of the RRI. Reverse remodeling was defined as a relative change in LVESVI of >15% [15].

Statistical analysis. Continuous variables are expressed as mean values ± standard deviation. Discrete variables were summarized as frequencies and percentages. Spearman analysis was used to evaluate the relationship between the changes in the RRI and changes in the other studied parameters. To study the effect of sacubitril/valsartan therapy, we applied a linear regression mixed model on the values obtained at the different time points (before and after the therapy), with patients fitted as subject-specific random intercepts. The effects of sacubitril/valsartan therapy and the interaction at different time points were considered. If the overall effect was significant in the linear model, then pairwise differences were examined. The trends over timing points were displayed by plotting the mean values with standard error. Statistical analyses were performed in STATA software, version 12 (StataCorp LLC, College Station, TX, USA) or Statistica 6.1 software (StatSoft Inc., Tulsa, OK, USA). A *p* value of <0.05 was considered statistically significant.

## 3. Results

A total of 80 consecutive patients for whom sacubitril/valsartan was prescribed were evaluated, and 14 patients were excluded as follows: eight because of sacubitril/valsartan intolerance (seven for hypotension and one for muscular pain) and six because of a missing T1 evaluation. The clinical characteristics of the remaining 66 patients are shown in Table 1.

As shown in Table 2, after sacubitril/valsartan administration, significant reverse remodeling was observed, i.e., a significant decrease in the left ventricular volumes, as well as significant improvements in the LVEF. The improvement of the left ventricular volumes and the LVEF were observed after the initiation of the sacubitril/valsartan. In patients in whom T-1 was available, no changes in the left ventricular volumes were observed when the T-1 and T0 evaluations were compared. The parameters reflecting the left ventricular filling pressures, i.e., E/e’, left atrial volume, and NT-proBNP, also significantly improved after sacubitril/valsartan, whereas no significant differences were found between the T-1 and T0 measurements. No significant differences were observed in TAPSE, TR, CVP, PAPs, creatinine, or GFR-EPI. 

When the RRI was analyzed, a significant reduction was demonstrated after the sacubitril/valsartan treatment, whereas no differences were found between the T-1 and T0 evaluations in the 41 patients in whom it was available (Figure 1, left panel).

In order to evaluate the relationship between the RRI changes and the improvements in the left ventricular remodeling and function, we separately evaluated the variation in the RRI in patients with and without reverse remodeling (i.e., a relative decrease in LVESV of >15%) (Figure 1, middle panel). Both patients with and without reverse remodeling showed an improvement of RRI when compared with baseline values. This improvement was even greater and more significant in the patients with reverse remodeling when a comparison with T-1 values was performed. Moreover, as shown in the right panel of Figure 1, no significant changes in RRI were observed according with sacubitril/valsartan dosage. 

Finally, the changes in the RRI were not correlated with those of the parameters reflecting the diastolic function, i.e., LAV (Spearman’s R 0.014, *p* 0.911) and E/e’ (Spearman’s R 0.116, *p* 0.391). Analogously, no correlation was found with the absolute changes in TAPSE (Spearman’s R −0.027, *p* 0.845) and PAP (Spearman’s R 0.008, *p* 0.956). In addition, the changes in the RRI were not correlated with the absolute and relative changes in creatinine (Spearman’s R 0.219 and 0.215, *p* 0.087 and 0.093, respectively) and with the relative changes in GFR (Spearman’s R −0.122, *p* 0.338). 

## 4. Discussion

The main finding of this study was that, after treatment with sacubitril/valsartan, significant improvements in renal resistances were observed and were not associated with the improvements in the left ventricular function.

Sacubitril/valsartan plays a key role in the treatment of patients with HFrEF [1,16]. The greater efficacy of this drug compared to ACEi [1,2] is related to the contemporary angiotensin II antagonism and the inhibition of neprilysin, the endothelial endopeptidase that is involved in the degradation of NPs. The NPS counteracts the renin-angiotensin system and sympathetic nervous system activity [5] by inducing natriuresis and diuresis, thus exerting an antifibrotic effect at the cardiac level, causing vasodilation and inhibiting the renin-angiotensin II system. These effects explain the improvement in cardiac function, which we observed in our series of patients. In fact, a significant decrease in left ventricular volumes, an improvement in the left ventricular systolic function, and a decrease in the left ventricular filling pressures were also observed in our patients.

However, the beneficial effects of sacubitril/valsartan are mediated not only by cardiac protection but also by renal protection, thus slowing the progression of cardiorenal syndrome [17] in patients with HF. The nephroprotective effects of sacubitril/valsartan are mediated by both antagonism of angiotensin II and by the inhibition of neprylisin [10] (Figure 2). The latter effect, by decreasing the degradation of NPs, increases cGMP-PKG activity, thereby leading to not only natriuretic and diuretic effects, but also other favorable effects, such as afferent arteriole dilation and increased glomerular filtration. Moreover, NPs inhibit sympathetic nervous system activity and angiotensin II, and consequently induce efferent arteriole dilation, glomerular hypertrophy, and scaring, as well as mesangial matrix accumulation. These effects explain the diminished renal fibrosis that is mediated by sacubitril/valsartan at the level of the kidneys [18].

Our results support the hypothesis that sacubitril/valsartan exerts specific nephroprotective effects that decrease arterial renal resistance and are associated with the progression of renal dysfunction. In fact, the RRI reflects arterial renal resistance [12,19,20], which is closely associated with the pathophysiology of renal dysfunction. The increased RRI is associated with the overactivation of the neuro-hormonal systems, as well as renal parenchymal abnormalities, leading to vascular rarefaction. The increased RRI is also associated with oxidative stress, endothelial dysfunction, and inflammatory cytokine activity [20,21]. Finally, in patients with heart failure, greater intrabdominal and central venous pressure can also increase the RRI, as a consequence of renal congestion [22]. Together, these mechanisms may explain the prognostic relevance of RRI, as well as the relationship between RRI and the progression of renal dysfunction. The ability of sacubitril/valsartan to decrease the renal resistance indicates that sacubitril/valsartan can modify the pathophysiological background underlying the RRI, thus providing renal protection.

Interestingly, the changes in the RRI that have we observed were not related to those reflecting reverse remodeling, improvements in LVEF, or decreased left ventricular filling pressure and right pressure. As a consequence, sacubitril/valsartan might be hypothesized to have additive and direct renal effects independent from the improvements in cardiac function. 

Our study did not include a control group. In order to overcome this limitation and to strengthen the evidence of a causal relationship between sacubitril/valsartan therapy and RRI variation, we evaluated the parameters not only at the time of prescription and after the treatment with sacubitril/valsartan, but also at least six months before the start of the treatment. Interestingly, the RRI, as well as the left ventricular atrial and ventricular volumes and the left ventricular ejection fraction, were similar at T0 and T-1, but changed significantly after the administration of sacubitril/valsartan. These findings provide further support for the drug’s role in the observed changes.

Finally, no relationship was found between the RRI changes and the changes in creatinine serum levels and estimated GFR. As previously demonstrated [19], an altered RRI may precede the decline in renal function, because it more accurately reflects the pathophysiological mechanisms leading to nephron loss, when a normal GFR is still present due to compensatory mechanisms.

However, our study presents several limitations. It was not a randomized study, and we did not have data to compare the effects of sacubitril/valsartan with those of ACE-inhibitors or angiotensin II receptor blockers. However, all of our patients were taking one of these two classes of drugs before sacubitril/valsartan and no change was observed between T-1 and T0 evaluations, whereas the changes were observed after the initiation of sacubitril/valsartan. The changes in RRI were statistically significant but small. This could be due to the short follow-up. However, it is worth noting that no significant changes in GFR were observed during the same period. Consequently, despite the above-mentioned limitations, our results could be useful to generate hypotheses about the effects of sacubitril/valsartan on renal resistances that could occur earlier than those of GFR. Future studies should confirm our results and evaluate the RRI changes in a larger population with a longer follow-up. 

## 5. Conclusions

Sacubitril/valsartan appears to have favorable effects on arterial renal resistances and kidney function. These variations are likely to be related to the combined renal effects of the inhibition of the renin-angiotensin system and the inhibition of neprylisin. Moreover, these effects are not related to reverse remodeling, changes in creatinine serum levels, or the estimated GFR. In this sense, more evidence is needed in order to better characterize the effects on renal function and hemodynamics.

## Figures and Tables

**Figure 1 jcm-11-03683-f001:**
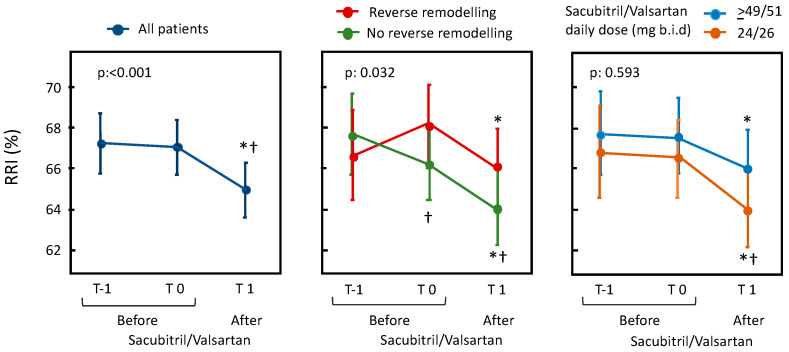
In the left panel, changes in RRI before and after sacubitril/valsartan are presented. In the right panels, changes are presented depending on the occurrence of reverse remodeling after treatment with sacubitril/valsartan and on its dosage. The data are expressed as the mean and 95% confidence interval with a linear mixed model adjusted for repeated measures. *p* refers to the statistical significance of the model; * *p* < 0.05 vs. T0, † *p* < 0.05 vs. T-1.

**Figure 2 jcm-11-03683-f002:**
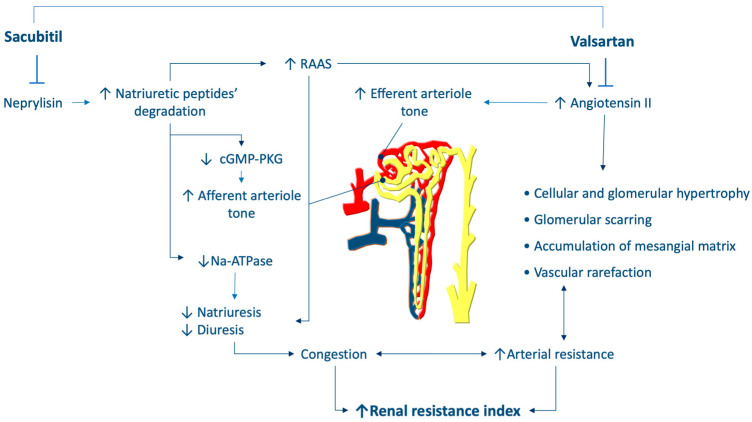
Hypothesis regarding the effects of sacubitril/valsartan on renal resistance. RAAS: renin-angiotensin system.

**Table 1 jcm-11-03683-t001:** Patient baseline clinical characteristics.

Number	66
Age (years)	56 ± 13
Males, *n* (%)	56 (85)
Ischemic etiology *n*, (%)	24 (36)
Diabetes mellitus *n*, (%)	13 (20)
Arterial Hypertension *n*, (%)	31 (47)
Atrial Fibrillation *n*, (%)	5 (8)
NYHA class II, *n* (%)	49 (76)
III, *n* (%)	17 (24)
BMI (kg/m^2^)	29.4 ± 6.2
SAP (mm Hg)	120 ± 15
Heart rate (beats/minute)	67 ± 9
LVEF (%)	29 ± 6
Creatinine (mg/dL)	0.99 ± 1.9
GFR-EPI (mL/min/1.73 m^2^)	84 ± 22
NT-proBNP (pg/mL)	1052 ± 1321
Concomitant therapy at the enrollment	
ACE-I, *n* (%)	45 (68)
Enalapril-equivalent dose (mg/die)	11 ± 6
ACE-I ≥ 50% target dose *n* (% among treated)	32 (71)
ARB, *n* (%)	21 (32)
Valsartan-equivalent dose (mg/die)	138 ± 75
ARB ≥ 50% target dose (% among treated)	11 (55)
Beta-blockers (%)	65 (98)
Bisoprolol-equivalent dose (mg/die)	7.1 ± 3.2
Beta-blocker ≥ 50% target dose	50 (76)
MRA *n*, (%)	58 (88)
MRA dose	45 ± 26
Loop diuretics *n*, (%)	52 (79)
Furosemide-equivalent dose (mg/die)	76 ± 102
ICD, *n* (%)	61 (95)
CRT, *n* (%)	22 (34)
Sacubitril/Valsartan up-titrated dose	
24/26 mg b.i.d., *n* (%)	34 (51)
49/51 mg b.i.d., *n* (%)	22 (34)
97/103 mg b.i.d., *n* (%)	10 (15)

ACE-I: inhibitors of Angiotensin-Converting Enzyme; ARB: angiotensin II receptor blockers; BMI: body mass index; GFR-EPI: estimated glomerular filtration rate by EPI formula; CRT: cardiac resynchronization therapy; ICD: implantable cardioverter-defibrillator; LVEF: left ventricular ejection fraction; MRA: mineralcorticoid receptor antagonists; NYHA class: New York heart Association class; NT-proBNP: amino terminal brain natriuretic peptide; SAP: systolic arterial pressure.

**Table 2 jcm-11-03683-t002:** Changes in studied parameters after sacubitril/valsartan treatment.

	Sacubitril/Valsartan			
	Before		After	
	T-1	T0	T1	*p*
SAP (mmHg)	122 ± 16	120 ± 15	116 ± 19 †	0.037
LVEDV (mL)	193 ± 50	184 ± 57	173 ± 56 *†	<0.001
LVESV (mL)	136 ± 41	133 ± 48	116 ± 46 *†	<0.001
LVEF (%)	30 ± 6	29 ± 6 †	34 ± 6 *†	<0.001
MR (a.u.)	1.8 ± 0.8	1.7 ± 0.8	1.6 ± 0.6	0.154
LAV (mL)	83 ± 29	82 ± 32	70 ± 27 *†	<0.001
E/e’	10.8 ± 3.4	10.9 ± 3.4	9.7 ± 3.9 *†	0.033
TAPSE (mm)	19.6 ± 3.8	19.8 ± 3.3	20.4 ± 3.4	0.281
TR (a.u.)	1.6 ± 0.7	1.5 ± 0.6	1.5 ± 0.6	0.541
CVP (mmHg)	4.9 ± 2.6	4.0 ± 2.2	4.4 ± 2.5	0.132
PAPs (mmHg)	32 ± 8	32 ± 7	30 ± 6 *	0.049
Creatinine (mg/dL)	0.96 ± 0.24	0.99 ± 0.26	1.01 ± 0.22	0.404
GFR-EPI (mL/min/1.73 m^2^)	87 ± 20	84 ± 21	83 ± 20	0.268
NTproBNP (pg/mL)	857 ± 1105	1052 ± 1321	614 ± 653 *†	0.017
RRI (%)	66.9 ± 5.5	67.0 ± 5.5	64.9 ± 5.5 *†	<0.001

Data expressed as mean ± standard deviation. *p* refers to linear fixed model. T-1 available in 49 patients. * *p* < 0.05 vs. T0; † *p* < 0.05 vs. T-1. CVP: central venous pressure; E/e’: the ratio between the peak of the E wave (E), through mitral pulsed Doppler at the level of the mitral leaflets, and early diastolic velocity peak (e’) at the level of the septal and lateral mitral annulus, through tissue Doppler imaging; GFR-EPI: estimated glomerular filtration rate by EPI formula; LAV: left atrial volume; LVEDV: left ventricular end-diastolic volume; LVEF: left ventricular ejection fraction; LVESV: left ventricular end-systolic volume; MR: mitral regurgitation; NT-proBNP: amino-terminal brain natriuretic peptide; PAPs: estimated systolic pulmonary arterial pressure; RRI: renal resistance index; SAP: systolic arterial pressure; TAPSE: peak of tricuspid annulus systolic excursion; TR: tricuspid regurgitation.

## Data Availability

The datasets used and/or analyzed during the current study are available from the corresponding author upon request.

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
