# Peer review of "Effects of Sacubitril/Valsartan on the Renal Resistance Index"

_jcm, 2022, doi:10.3390/jcm11133683_

Round 1
Reviewer 1 Report
Gioia M.I. and col. analyzed the effects of sacubitril/valsartan treatment on renal resistance index (RRI) measured by doppler ultrasound in a group of patients with chronic heart failure. The authors found a significant decrease in RRI at the six-month follow-up, suggesting beneficial effects of the drug combination on renal perfusion, independent of cardiac changes.
I have minor comments:
1. In methods, measurement of renal resistance index is briefly described, with the authors sending the reader to another paper. Nevertheless, I think it should be more extensively presented.
2. Was the renal ultrasound done by a single examiner? How did the authors prevent excessive measurement variability?
3. It is not clear how many Doppler measurements were done. How many cardiac cycles? In atrial fibrillation patients, more cycles were averaged? How many per kidney? Also, are the RRI values presented a mean of the values for the two kidneys?
4. Given that the observed decrease in RRI was rather small (approx. 2%), it would have been preferable to include a greater number of patients or a control group. Are the authors confident that the results are not by chance?
Also, as a side note, it would be interesting to see a comparison to valsartan or an ACE inhibitor.
Author Response
Reviewer 1
We would like to thank the reviewer for his/her helpful comments. This is our point to point reply.
- In methods, measurement of renal resistance index is briefly described, with the authors sending the reader to another paper. Nevertheless, I think it should be more extensively presented.
Response.
We thank the reviewer for the suggestion. The method was further clarified by adding further information at lines 91-106:
The method to assess RRI was already described previously [12,13]. Renal arterial Doppler was performed after echocardiographic examination by using the same echograph (Vivid 7, GE Vingmed Ultrasound, General Electric or EPIQ CVx system, Philips, Amsterdam, the Netherlands) and the same 4 MHz probe, moving the patient into the sit-ting position and using a posterior approach to the kidney. The course of right or left kidney segmental arteries was visualized by color Doppler flow and then, at the middle tract level of the best one visualized, pulsed Doppler was performed. Every effort was made to achieve the best alignment of the ultrasonic beam. An average of 2–3 measurements of the peak systolic velocity and the end-diastolic velocity were used to calculate the RRI according to Peurcelot's formula, i.e. 100 × [1 – (end-diastolic velocity/peak systolic velocity)].
- Was the renal ultrasound done by a single examiner? How did the authors prevent excessive measurement variability?
Response.
We added information about the examiner at lines 102-105:
RRI is a parameter with a high inter-operator and intra-operator reproducibility as pre-viously demonstrated [12]. To avoid bias in the measurement, the images were acquired by a single operator (M.I.) and analyzed by a single operator for each center (M.I.G. for patients referred to the center of Bari and G.A. for those referred to the center of Foggia).
The measurement variability was already evaluated in a previous study of our group of research. RRI showed a high intra- and inter-operator reproducibility (see ref #12). We added this information in the methods section.
We added this information
- It is not clear how many Doppler measurements were done. How many cardiac cycles? In atrial fibrillation patients, more cycles were averaged? How many per kidney? Also, are the RRI values presented a mean of the values for the two kidneys?
Response.
We added information about this in the methods section (see above).
Given that the observed decrease in RRI was rather small (approx. 2%), it would have been preferable to include a greater number of patients or a control group. Are the authors confident that the results are not by chance?
Response.
Our study was based on a relatively short follow-up. This can justify the small change of RRI, which, however, was highly significant. Moreover, if our results were by chance, differences between T-1 and T0 would have been present.
However, we reported the small observed effect as possible limitation of the study at lines 244-255:
Limitations of the study. Our study presents several limitations. It was not a randomized study and we have not data to compare the effects of sacubitril/valsartan with those of ACE-inhibitors or Angiotensin II receptor blockers. However, all our patients were taking one of this two classes of drugs before sacubitril/valsartan and no change was observed between T-1 and T0 evaluations, whereas the changes were observed after the initiation of sacubitril/valsartan. The changes in RRI were statistically significant but small. This could be due to the short follow-up. However, it is worth noting that no significant changes of GFR were observed during the same period. Consequently, although with the above mentioned limitations, our results could be useful to generate hypotheses about the effects of sacubitril/valsartan on renal resistances which could occur earlier than those of GFR. Future studies should confirm our results and evaluate RRI changes in a larger population during a longer follow-up.
Also, as a side note, it would be interesting to see a comparison to valsartan or an ACE inhibitor.
Response.
This is a very interesting point. We have no data to compare the effects of valsartan or an ACE-inhibitor. We stated this in a section of limitations of the study (see above).

Reviewer 2 Report
Authors presenting an interesting study that analyze the RRI changes after S/V initiation in patients with stable HFrEF NYHA II-IV.
The design of the study is appropriate as hypothesis generator study. However, I have some suggestions about the manuscript:
1.- Authors must specified better the primary and secondary endpoints of the study in a specific paragraph inside of Materials and Methods.
2.- RRI evaluation: Is doppler evaluation the gold standard technique to evaluate the RRI? There are better methods? Please specified it.
3.- Table 1 (Baseline clinical characteristics): I think that authors presents here clinical data of T0. Although the NT-proBNP levels are quite similar to T0 (1050 vs 1052), please modified it. Both value should be equals.
4.- An absolute reduction of 2% since a slight numerical reduction. However the p value is very significant. ¿Could you specified better the statistical analysis or could you send me the statistical analysis to check the result?
5.- ¿How and why authors decided that a significant reverse remodeling was a decrease of LVESV > 15%?
6.- Conclusions: Authors should eliminate the second phrase (these effects.....natriuretic peptides) because is an opinion. They should add the absence of relationship between echocardiogram and renal function parameters (GFR and creatinine) changes.
I have doubts about the p value of the primary endpoint. Authors report a reduction of 2% in RRI value, with a p value < 0.001. I don't know if is possible to check this statistical analysis with an specific person.
Author Response
see the file attached.

Round 2
Reviewer 2 Report
Agree with the modifications and the explanations.
Author Response
Thank you for your comment.